# CFD Modelling of Gas-Solid Reactions: Analysis of Iron and Manganese Oxides Reduction with Hydrogen

Mopeli Khama [1,*] and Quinn Reynolds [1,2]

[1] Mintek, 200 Malibongwe, Randburg 2194, South Africa; quinnr@mintek.co.za
[2] Department of Chemical Engineering, University of Stellenbosch, Private Bag X1, Matieland 7602, South Africa
[*] Correspondence: mopelik@mintek.co.za

**Abstract:** Metallurgical processes are characterized by a complex interplay of heat and mass transfer, momentum transfer, and reaction kinetics, and these interactions play a crucial role in reactor performance. Integrating chemistry and transport results in stiff and non-linear equations and longer time and length scales, which ultimately leads to a high computational expense. The current study employs the OpenFOAM solver based on a fictitious domain method to analyze gas-solid reactions in a porous medium using hydrogen as a reducing agent. The reduction of oxides with hydrogen involves the hierarchical phenomena that influence the reaction rates at various temporal and spatial scales; thus, multi-scale models are needed to bridge the length scale from micro-scale to macro-scale accurately. As a first step towards developing such capabilities, the current study analyses OpenFOAM reacting flow methods in cases related to hydrogen reduction of iron and manganese oxides. Since reduction of the oxides of interest with hydrogen requires significant modifications to the current industrial processes, this model can aid in the design and optimization. The model was verified against experimental data and the dynamic features of the porous medium observed as the reaction progresses is well captured by the model.

**Keywords:** OpenFOAM; gas-solid reactions; fictitious domain; hydrogen reduction

## 1. Introduction

Iron and steel have a significant economic importance, therefore the reduction of iron oxides is of interest to many researchers. There is a high carbon footprint from steel and iron making industries, and it is crucial to develop technologies on renewable energy to address the high $CO_2$ emissions. This is because crude steel is primarily produced from blast furnaces which utilize carbon as both a source of energy and a reducing agent and as a result the $CO_2$ emissions are high [1,2]. $H_2$ as a reducing agent and a source of energy is a promising route to replace carbon if $H_2$ is produced from renewable sources. It presents many advantages as a reductant in that there is better contact between the gas and the solid in comparison to solid-solid reduction [3]. The low viscosity, high mobility and small molecule size promote faster diffusion of $H_2$ through the pores and ultimately faster reaction rates in comparison to other gaseous reductants such as CO and $CH_4$ [4].

However, the use of $H_2$ as both the source of energy and a reducing agent still faces significant challenges. The challenges include the inability to produce the needed $H_2$ from the renewable energy and the substitution of coke with $H_2$. The latter challenge is compounded by the endothermic nature of $H_2$, which necessitates a need to add coke for energy. The disadvantages of using $H_2$ as a reducing agent is that it has a lower affinity to oxygen than carbon and this renders it unsuitable for the reduction of ignoble metals [5]. Furthermore, reduction and pre-reduction of ores by carbon is thermodynamically feasible at high temperatures, while $H_2$ can only be used for pre-reduction and reduction of certain ores. To circumvent this limitation in using $H_2$ as a reducing agent for a wide range of ores, low partial pressure ratio of $H_2O$ to $H_2$ is used.

The reduction of haematite ($Fe_2O_3$) into iron (Fe) is complicated by structural changes in the intermediate oxides and diffusion in the pores which limits the access of the reducing agent. Pore structure has a significant influence on the overall process rates. The reduction of iron ores by gaseous reducing agent is a typical example of heterogeneous reactions whose rates are affected by both chemical kinetics and heat and mass transfer factors. A heterogeneous reacting flow model that takes into account the reactions in the porous medium coupled with momentum and heat transfer would be of benefit in helping to optimize these systems. The heterogeneous reactions are dynamic in space and time and result in the formation of surface layers, changes in the composition of the solid including some structural changes [6]. These are crucial features that require to be captured by a numerical model. In addition, the reduction with $H_2$ requires significant changes to the current production facilities, therefore, there is a need to develop CFD tools that will aid the design and optimization. Such CFD models need to be comprehensive and computationally efficient.

The $H_2$ reduction of oxides is a dynamic process and the reaction rates depend on pressure, temperature, porosity, particle size, mineralogy and gas composition etc. Furthermore, these types of reactions are characterized by a hierarchy of phenomena that influences the reactions at different length and time scales. This hierarchy of phenomena includes macroscopic scale processes such as transport and reaction kinetics in a reactor [4]. Multiscale simulations of $H_2$ reduction of oxides will play a crucial role in bridging the length scale from atomic scale where elementary red-ox reactions take place up to the reactor level where macroscopic gas transport and gradients in the local boundary conditions become relevant. The development of a model that is able to capture the hierarchical phenomena from micro-scale to macro-scale will benefit process development in the use of $H_2$ as a reducing agent.

Several authors have developed models for studies on gas-solid reactions. Ref. [7] developed a 2D model for the direct reduction of iron oxide in a shaft furnace in the steady-state regime. Their model adopts a four scale approach that includes the bed of iron pellets, the iron pellets, and the grains and the crystallites that make up the grains. The governing equations are solved iteratively by employing the finite volume method and the kinetics solved from a single pellet kinetic sub-model. Their findings indicate that at 700 °C, Fe degree of metallization is 70% and there exist strong radial gradients. On the other hand, from 900 to 950 °C, Fe metallization of 100% was achieved with flatter radial gradients.

CFD-DEM models have also been utilized for the modelling of gas-solid reactions [8]. A CFD multi-scale modelling for the catalytic gas-solid reactions was used by [9] to analyse the macroscopic behaviour of the reactive fluidized bed reactor. The authors indicate that the macroscopic behaviour is a result of the coupling between the different spatial and temporal scales. In their work, the reactivity of the system is governed by the elementary reactions at the atomic level, while the species distribution in the reactor is governed by fluid dynamics. This hierarchy of phenomena is also observed in the $H_2$ reduction of iron oxides.

The coupling of fluid flow and chemistry in fixed bed reactors has in the past been modelled by making an assumption of plug flow and effective transport mechanisms in the reactor. The limitation of models that assume a plug flow was addressed by [10]. In their model, a 2D CFD heterogeneous reactor model that uses the one-equation turbulence to describe the turbulence in the fixed bed reactor is utilized. Their findings indicate that by employing a two temperature porous medium model, a close agreement between experimental and model predictions is found.

A standalone CFD solver that couples fluid flow, heat transfer and heterogeneous reactions was developed to model catalytic reactions by [11]. This micro kinetics solver allows for multiscale modelling for general geometries and it is an efficient solver since it uses the operator splitting methods and avoids the costly matrix operations which are often encountered in fully coupled approaches. As a result, the detailed reaction mechanism are modelled at moderate computational expense and this captures the chemistry accurately.

Ref. [12] developed a CFD code for coupling kinetics and fluid flow in a porous fixed bed reactor for both transient and laminar flows. The plug flow assumption is relaxed and radial gradients can be estimated. The porous medium is modelled as a field inside the computational domain defined by its physical properties and the heterogeneous reactions are defined as a volume phenomenon.

The purpose of the current study is to employ the CFD models to analyse the $H_2$ reduction of iron ore by considering gas-solid reactions in the porous medium and the influence of flow field on the reactions rates. Diffusion resistance is a common problem in gas-solid reactions and the study incorporated this phenomena which helped predict the operating regime. The mass and heat transfer from the gas phase to the solid porous medium has also been taken into account.

## 2. Numerical Model

The modelling of gas-solid reactions should be comprehensive enough to take into account the structural changes of the porous medium during the reaction and the transport of species into the porous medium. Furthermore, heat transfer from both the solid and gas phase and the evolution of heat as a result of the reactions should also be taken into consideration. The knowledge of heat transfer in the porous medium of great importance particularly in reaction kinetics as this helps the prediction of the variation in reaction rates due to among other factors the disturbances in the inlet temperatures. This is crucial as the rate of iron ore reduction is dependent on the reduction temperature. The pressure drop in the porous medium is taken into account by the introduction of the Darcy's term in the Navier-Stokes equation as represented in Equation (7).

Gas utilization is an important factor in the viability of $H_2$ reduction reactions and the numerical models are important tools in helping to determine the operating conditions that allow for optimum $H_2$ utilization [13]. Reduction by $H_2$ is endothermic, therefore external heating is required. In the current study, the heating rate of 5 °C/min was used for heating until the pre-determined temperature of 900 °C was reached and maintained. The heating rate was implemented as a time dependent boundary condition for temperature.

### 2.1. The Reaction System

The reaction system constitutes porous iron ore pellets that are reduced by $H_2$ or a mixture of $H_2$ and CO. However, the introduction of CO inhibits the reduction reaction since $H_2O$ cannot be removed from the reaction zone due to the equilibrium water-gas shift reactions. The ratio of the concentration of $H_2O$ to $H_2$ needs to be low for faster reduction reactions. The gases flow from the bulk gas phase to the porous medium through convection and diffusion and the reactions take place in the porous medium. The gaseous products flow by diffusion through the porous medium to the gas phase. The counter-diffusive flow of products and reactants through the porous medium affects the rate of heterogeneous reactions. Since the gases flow by diffusion through the pores, there is a possibility of diffusion limited reactions and a numerical model should capture this phenomenon. In particular, the pellets with low porosity and gases that flow at low velocity are more susceptible to diffusion limitations. These phenomenon is well captured by the reacting flow model used in the current study.

The reduction of $Fe_2O_3$ by $H_2$ is a multi-step reaction system as represented by Equations (1)–(4) [14].

$$3\,Fe_2O_3 + H_2 \rightarrow 2\,Fe_3O_4 + H_2O \tag{1}$$

$$Fe_3O_4 + 4\,H_2 \rightarrow 3\,Fe + 4\,H_2O \tag{2}$$

$$Fe_3O_4 + H_2 \rightarrow 3\,FeO + H_2O \tag{3}$$

$$FeO + H_2 \rightarrow Fe + H_2O \tag{4}$$

Unlike the reduction of iron oxides which results in full metallization, the reduction of manganese ores by $H_2$ does not result in total reduction, instead the higher order oxides are reduced to MnO. The reaction mechanism by [15] is shown in Equations (5) and (6).

$$3\,MnO_2 + 2\,H_2 \rightarrow Mn_3O_4 + 2\,H_2O \tag{5}$$

$$Mn_3O_4 + H_2 \rightarrow 3\,MnO + H_2O \tag{6}$$

The reaction system constitutes a series of intermediate reactions and the governing equations have to be solved for each reaction step which results in increased computational expense. In order to address the high computational expense, several methods such as a reduced mechanism are often used. Ref. [12] developed a heterogeneous reacting flow CFD model based on the special type of immersed boundary (fictitious domain) to predict reaction rates in porous medium. The solver couples the transport phenomenon with reaction kinetics and was used in the current study.

### 2.2. Governing Equations

The gas-phase is described in the Eulerian framework and governing continuity and momentum equations are shown in Equations (7) and (8). The source term (R) resulting from the gas-solid reactions and the heat source term ensure the coupling between the gas and solid phases [16]. The energy balance equations are described by Equations (10) and (13) and the coupling between the phases is done through the heat source term.

$$\frac{\partial}{\partial t}\rho^G u + \nabla.\left(\rho^G\,uu\right) + \varepsilon\nabla P - \nabla.\left(\mu_{eff}\nabla u\right) - \rho^G g = -\mu_{eff}D.u - F.u \tag{7}$$

$$\frac{\partial}{\partial t}\varepsilon\rho^G + \nabla.(\rho^G u) = (1-\varepsilon)\sum_i R_i^{Reduction} \tag{8}$$

$$\frac{\partial}{\partial t}\varepsilon\rho^G y^G + \nabla\left(\rho^G\,uy^G\right) - \nabla.\left(\varepsilon\rho^G D_{eff}\nabla Y^G\right) = \varepsilon w^G + (1-\varepsilon)R_i^{Reduction} \tag{9}$$

$$\frac{\partial\left(\varepsilon\rho^G Cp^G T^G\right)}{\partial t} + \nabla.(\rho^G Cp^G T^G u) - \nabla.\nabla\left(\varepsilon k_{eff}^G T^G\right) = -\varepsilon\sum_i w_i h_{f,k}^0 - h_{conv}S_{Av}\left(T^G - T^S\right) +$$
$$(1-\varepsilon)T^G \sum_i Cp_i\,R_i^{Reduction} + S^{G,Radiation} \tag{10}$$

In the Equations (7)–(10), $G$ denotes the gas phase, $\epsilon$ is porosity, $\rho$ denotes density, $k_{eff}$ is the gas thermal conductivity, $D$ is Darcy's resistance to flow, $D_{eff}$ is the diffusion coefficient, $\mu_{eff}$ is the dynamic viscosity, $u$ is the gas velocity, $C_p$ is the specific heat. The governing equations for the solid phase constitute the continuity, species conversion and energy balance as represented by Equations (11)–(13).

$$\frac{\partial}{\partial t}(1-\varepsilon)\rho^S = (1-\varepsilon)\sum_k R_K^{Reduction} \tag{11}$$

$$\frac{\partial}{\partial t}(1-\varepsilon)Y_k^s\rho^S = (1-\varepsilon)\sum_k R_K^{Reduction} \tag{12}$$

$$\frac{\partial}{\partial t}(1-\varepsilon)\left(\rho^S C^S T^S\right) - \nabla.\left((1-\varepsilon)\mathbb{K}k_{eff}^s.\nabla T^S\right) = h_{conv}S_{Av}\left(T^G - T^S\right) + Hr + S^{S,Radiation} \tag{13}$$

In Equations (11) and (12), $Y$ denotes the solid species mass fraction, $\mathbb{K}$ is the anisotropy, $k_{eff}^s$ is the conductivity coefficient, $Hr$ denotes the heat of reaction which is calculated as presented in Equation (14).

$$Hr = (1-\varepsilon)\sum_k h_{f,k}^0 R_k^{Reduction} - (1-\varepsilon)\sum_i Cp_i T^G R_i^{Reduction} \tag{14}$$

In Equation (14), $R_k$ is the reduction rate of the solid components while $R_i$ is the reduction rate of of the gas species.

The Naiver-Stokes equations are solved only in the fluid domain, but the interface conditions on the boundary of the porous material are taken into account. The derivation and solution of the momentum equations is performed according to the porous medium theory wherein permeability relative to each medium is introduced. In this case, the velocity and pressure inside the porous medium is calculated.

Equations (7)–(13) are non-linear and coupled and the physical and chemical properties changes as the reactions take place, therefore, there are no analytical solutions. The finite volume methods are used in the discretization and solution of the governing equations. The porous Gasification Foam solver developed by [12] was coupled with OpenFOAM 8 to effect the coupling of chemistry and transport for the modelling of gas-solid reactions. Since the governing equations are non-linear and stiff with complex chemistry, the seulex ODE solver was used because it is robust and efficient when solving such systems. Although the seulex ODE solver is efficient, the backward difference formular (BDF) methods are extremely computationally efficient in the time integration of stiff reactions. This is because the computational effort is concentrated at the beginning of the integration and moving to the next time, the information from the previous is required and it is usually stored. The current study used the extrapolation methods for time integration of the stiff equations instead of BDF, but BDF have been reported to be extremely efficient in modelling combustion reactions with detailed mechanisms.

### 2.3. Boundary Conditions

The no slip boundary condition is defined for the velocity at the walls and at the inlet, velocity is specified. At the outlet, the zero gradient for velocity in defined and a fixed value of pressure is assigned. The coupling reaction between the gas and solid phases is endothermic and the pre-heated gas is injected into the reactor. The gas is heated at a rate of 5 °C/min until the required reduction temperature reached and thereafter the temperature is kept constant. The convective heat transfer at the porous medium interface is defined by Equation (15).

$$q = S_{AV} h (T_g - T_s) \tag{15}$$

where $S_{AV}$ is the surface area to volume ratio, $h$ is the convective heat transfer coefficient, $T_g$ the gas phase temperature and $T_s$ is the solid phase temperature.

The conductive heat flux at the porous medium interface is taken to be zero as described by Equation (16).

$$k \frac{\partial T}{\partial R} = 0 \tag{16}$$

### 2.4. Reaction Kinetics

The reactions concern the heterogeneous reactions which involve an interface between a gas and solid phase and the reactions occur on the surface of the adsorbed layer. The reaction kinetic models are coupled with the transport equations as demonstrated in the governing equations (Equations (7)–(14)). The chemical equations governing the evolution of the reduction process are non-linear and stiff, and thus introduce high computational cost in the solution of the governing equations. The reaction rate for the consumption and formation of gas phase and solid phase species are presented in Equation (17). The reaction rate source term in Equations (11), (12) and (14) is defined in Equation (17).

$$R_i^{Reduction} = \sum_{k=1}^{ks} v_{i,k} K_{f,i} \rho^s \prod_{j=1}^{Ng+Ns} Y^{v'_{j,k}}, \ i = 1, 2, \ldots, N_g + N_s \tag{17}$$

where $v_{i,k}$ is the stoichiometric coefficient, $Y$ is the species mass fraction, $k_s$ is the number of elementary reactions, $N_g$ is the number of gas phase species, $N_s$ is the number of solid

phase species. $K_f$ is the rate constant and $v'_{j,k}$ is reaction order. The rate constant depends on temperature and is described by Equation (18).

$$K_f = A exp\left(-\frac{Ta}{T}\right) \tag{18}$$

where $T_a$ is the activation temperature and $A$ is the pre-exponential factor.

The kinetic data for calculating the rates of reactions described in Equation (17) are obtained experimentally in the absence of heat and mass transfer limitations. The intrinsic rates are used to avoid the double inclusion of the diffusion terms, as the diffusion terms are included in the overall reactor model [17]. The spatial and temporal scale involved in solving these systems are wide which exacerbates the numerical efficiency. Furthermore, in cases where there is a large number of species and a detailed reaction mechanism, the computational cost is significantly high even for ideal systems. A significant amount of computation time in reacting flows goes to the evaluation of the Jacobian matrices. The Jacobian matrices are used in the computation of reactions rates and evaluating thermodynamics properties, therefore with a detailed reaction mechanism, the computation time increases significantly.

*2.5. The Inclusion of Pellets into the Computational Domain*

The inclusion of pellets into the computational domain is done by using the special form of the immersed boundary method named the fictitious domain. The immersed boundary method is used to include stationary or moving bodies into the spatial discretization without having to use the body conforming mesh [18]. The method concerns the immersing of the original physical boundary into a simply-shaped domain named fictitious domain. It allows for the use of structural meshes since the spatial discretization is performed on the fictitious domain. The strength of this approach lies in the ability to use fast solvers for the auxiliary problem on complicated geometries. When performing discretization, simple cartesian meshes are used and mesh grid is disconnected from the geometry of the sub-domains. A single rectangular cartesian grid is projected over the entire fictitious domain independent of the shapes of the porous material [19]. The geometry of the porous medium is taken into account by its physical characteristics. In the present study, the porous medium in the fictitious domain is taken into account by its physical characteristics which include porosity and viscous resistance. This is an efficient and easy implementation which allows for fast numerical computation particularly for cases where there is need to change the physical characteristics of the geometry or flow during calculation time without re-meshing the domain and re-writing the corresponding boundary conditions. In essence, the iron pellets are modelled as a collection of scalar and tensor fields such as thermal conductivity of the porous medium, density, viscous resistance and mass fraction.

## 3. Geometry and Meshing

The direct reduction or pre-reduction of iron ores is often carried out in a shaft furnace. The current study used the geometry shown in Figure 1 for the gas-solid reactions. The 2d axisymmetric geometry and a full 3D geometry were both investigated to gauze the effect of axisymmetry in the accuracy of the reduction reaction kinetic modelling. A grid convergence index for mesh refinement developed by [20] was employed to determine the error band and order of convergence on grid convergence of the numerical solution. The mesh sensitivity study was carried out using the Rachardon's interpolation method to determine the optimum grid size that results in resolution independent results. As observed in Figure 2, the temperature profiles for grid 1 and 2 are similar, whereas grid 3 temperature profile is different. The refinement analysis revealed that the asymptotic range is achieved and this confirmed that an optimum mesh size was chosen.

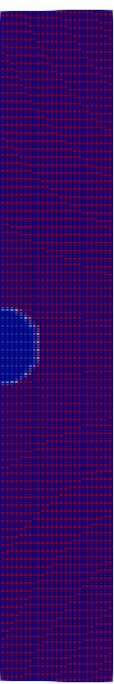

**Figure 1.** 2D axisymmetric geometry with a spherical pellet.

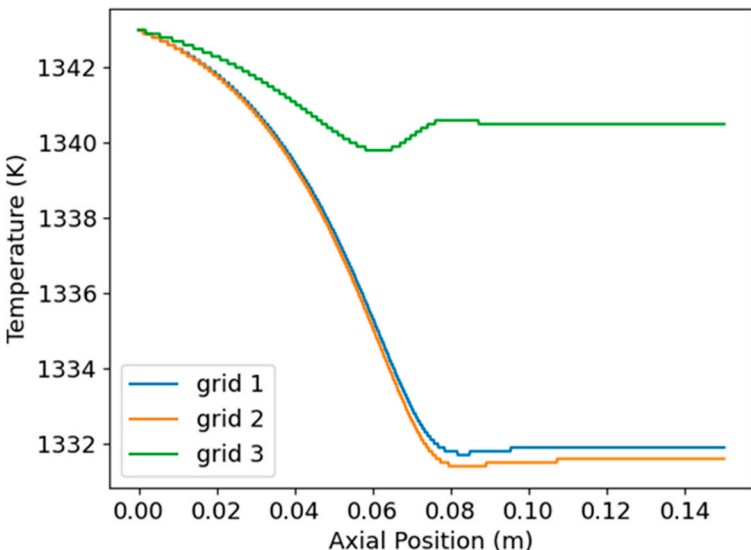

**Figure 2.** Temperature profiles at various computational grids.

## 4. Modelling a Single Pellet

Different scales of porosity can be considered in the modelling of heterogeneous gas-solid reactions of $H_2$ with iron pellets. In the first porosity scale, a single pellet of either a spherical or cylindrical shape is considered and the flow around and within the particle coupled with reactions is modelled. At this scale, the computational expense is moderate. The second approach is the modelling of a collection of pellets with a uniform or non-uniform porosity across the space they occupy. At this scale, the computational expense is high and the effects of diffusion limitations on the reaction rates is pronounced. The current study focuses on the flow within and around a single pellet. The graphical representation of the scale considered is shown in Figure 3. The computational grid is refined to a second level of refinement only where the gradients exist, that is the immediate vicinity of the particle. In this region, there is heat transfer between the gas and solid phase and the reactions take place in the porous medium.

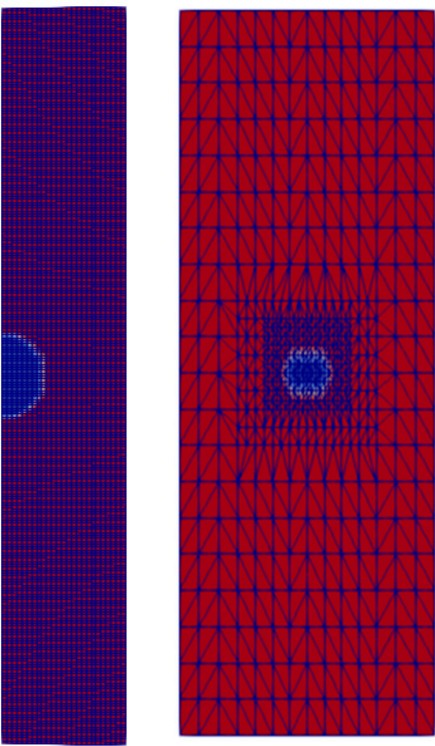

**Figure 3.** A single pellet included in a computational domain.

## 5. Results and Discussions

The reduction of iron oxide by $H_2$ in a single spherical particle was studied and this is a case of a meso-scale approach. At this scale, the diffusion path length is shorter compared to the macro-scale and it is expected that the diffusion will not play a significant role in inhibiting the reaction rates. The gas flowrate was 2 L/min and porosity of the pellet 0.33. The results in Figures 4 and 5 show the mass fraction profiles of reactants, intermediate products and products in an 8.5 mm spherical pellet. As revealed from the figures, at 1000 s, the full metallization is achieved. The intermediate oxides ($Fe_3O_4$ and FeO) are also fully converted at the final simulation time and the mole fraction of Fe is unity which reveal total reduction of iron oxides by $H_2$.

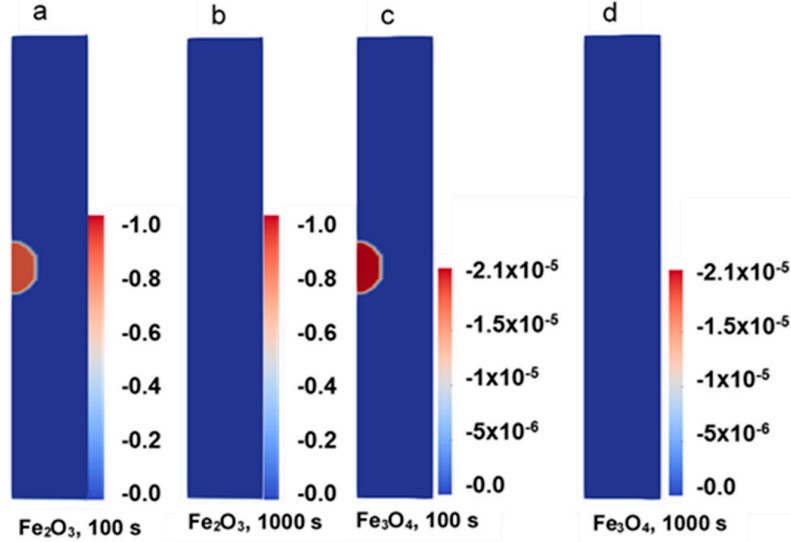

**Figure 4.** Mass fraction profiles for $Fe_2O_3$ (**a**,**b**) and $Fe_3O_4$(**c**,**d**) at reduction times of 100 and 1000 s.

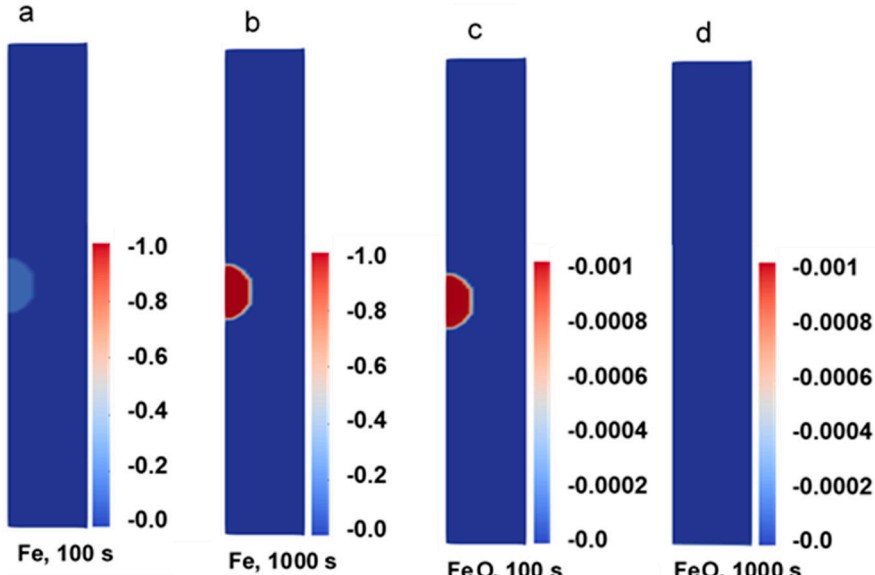

**Figure 5.** Mass fraction profiles for FeO (**c**,**d**) and Fe (**a**,**b**) at reduction times of 100 and 1000 s.

The high ratio of $H_2$ to $H_2O$ is required to facilitate higher degrees of reduction. The heterogeneous reactions in the porous medium result in mass addition to the continuous phases in the form of $H_2O$. As a result, the counter diffusion (diffusion of $H_2O$ from the pores to the gas phase and diffusion of $H_2$ from the gas phase into the pores) may serve as a barrier for the access of $H_2$ to the reaction site. The mass fraction profiles from the heterogeneous reactions are influenced by fluid flowrate, species concentration, reduction temperature and porosity. The mass fraction profiles in Figure 6 reveal that the ratio of $H_2$ and $H_2O$ is always high and this meets the criterion required to achieve higher degrees of metallization. At 1000 s, the reactions have reached completion and as a result, there is no $H_2O$ in the pellet and the surrounding flow field as shown in Figure 6d.

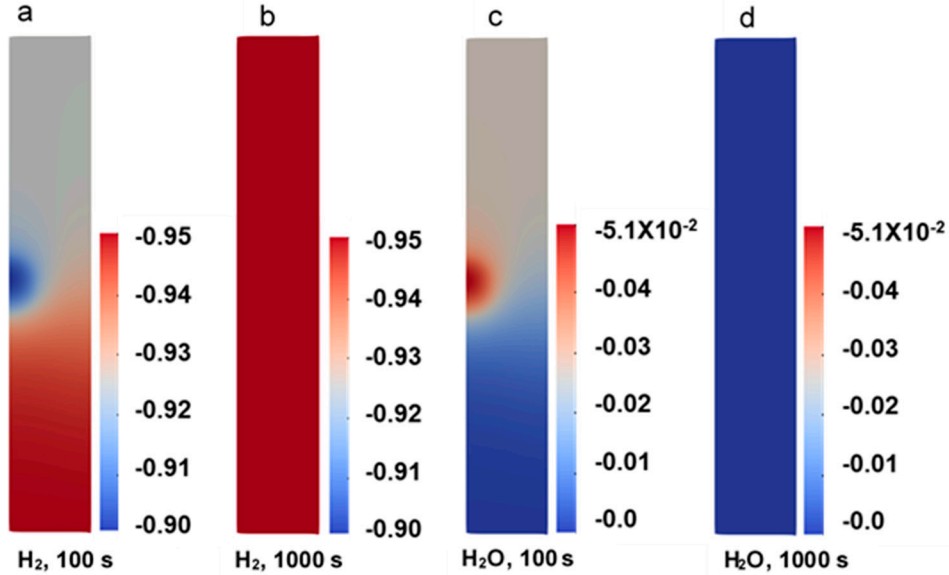

**Figure 6.** Mass fraction profiles of $H_2$ (**a**,**b**) and $H_2O$ (**c**,**d**) at reduction times of 100 and 1000 s.

The results in Figures 5 and 6 indicate that it takes a longer time to reach a complete Fe metallization. This can be ascribed to either diffusion or kinetic limitations. One way to determine whether reaction rates are controlled by diffusion or kinetics is to compare the species mass fraction at the particle surface and the center. If the mass fraction at the surface and the center are different, then diffusion limits the reaction rates. On the other hand, if

the surface and center mass fraction are the same, diffusion is faster than the rate of reaction and reaction rate are under kinetic limitations. The results in Figure 7 depict the mass fraction profiles of $H_2$ and $H_2O$ at particle's radial coordinate. The mass fraction profiles were taken from the cross section of the geometry shown in Figure 8. As observed, the $Fe_2O_3$ surface and center mass fractions are significantly different rendering the operating regime under diffusion limitations. On the other hand, $H_2$ and $H_2O$ mass fractions show that the mass fraction of $H_2$ is lower at the center of the pellet compared to the surface indicating the possibility of diffusion limitations.

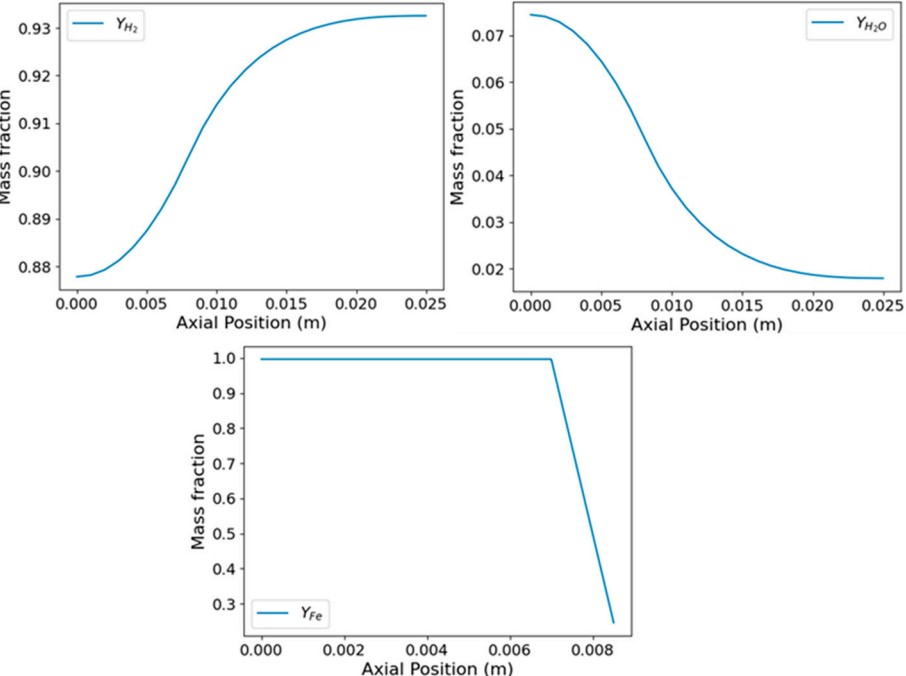

**Figure 7.** $Fe_2O_3$, $H_2$ and $H_2O$ mass fraction profiles along the x axial coordinate.

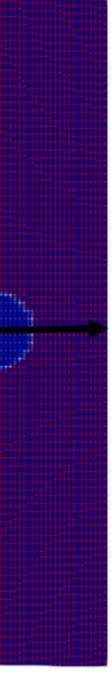

**Figure 8.** The geometric cross section depicting the region where the profiles were extracted.

A more robust approach to determine the operating regime is using the ratio of the reaction rate flux to the diffusion flux. If the ratio is greater than unity, the reaction rates are controlled by diffusion. As observed in Figure 9, the ratio of the reaction rate flux to diffusion flux is always greater than unity, with the ratio highest at the center of the pellet and decreasing towards the surface where the effect of diffusion limitations on the reaction rates is minimal.

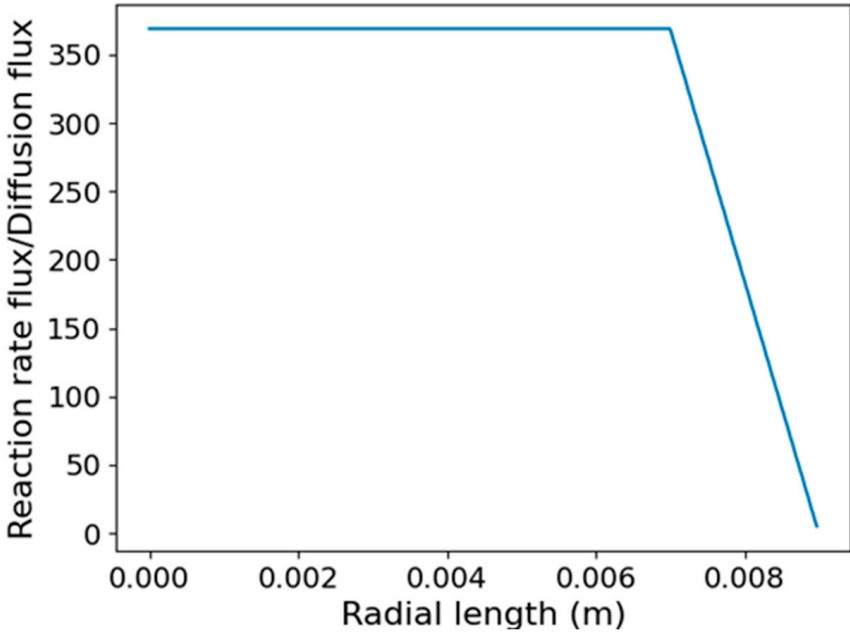

**Figure 9.** The ratio of reaction rate to diffusion flux along the pellet radial coordinate.

The flow within and around the porous medium is taken into account and the pressure drop is predicted from the Darcy's resistance equation inclusion into the momentum equation. The velocity profiles around the spherical particle and pressure drop is shown in Figure 10. The velocity profiles reveal the resistance to flow phenomenon inside the porous medium as shown the by streamlines. The velocity magnitude near the spherical particle is less than the velocity in the free stream region and the residence time in this region is longer. The longer residence time coupled with the high solid temperature facilitate the reduction process to full metallization.

The $H_2$ reduction of iron oxide is an endothermic process and requires the gas to be pre-heated to facilitate the reactions. The hot gas exchanges energy with the solid phase until thermal equilibrium between the two phases is reached. The temperature profiles are presented in Figure 11. At the end of the simulation time, thermal equilibrium between the solid and gas is reached as observed in Figure 11c,d. The radial gradients within the pellet are steep and the assumption of thermal equilibrium would not be valid. The variation of the temperature profiles along the pellet's radial coordinate are presented in Figure 12. As observed, the gradients are steeper in a 14 mm pellet in comparison to the 8.5 mm pellet.

### 5.1. Reduction of Manganese Oxides

The reduction of manganese oxides with $H_2$ does not result in complete reduction, but reduces to MnO as observed in Figure 13. The kinetics are also slower in the case compared to the reduction of iron oxides. The reduction to MnO only gets to 78% degree of reduction even for simulation times longer than iron oxide reduction at the same conditions. The reason for slower kinetics is attributed to higher activation energies in the reduction of low grade manganese ores used in the current study.

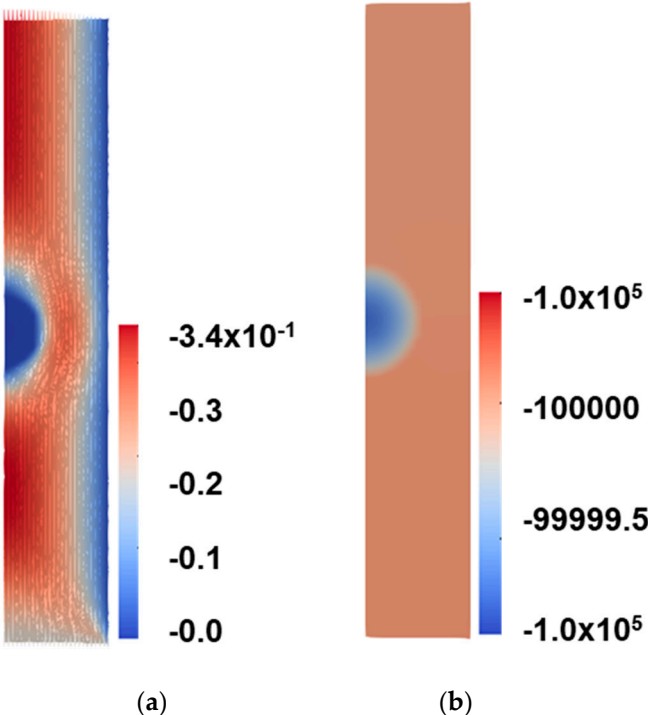

**Figure 10.** The velocity (**a**) and pressure (**b**) profile around a single pellet.

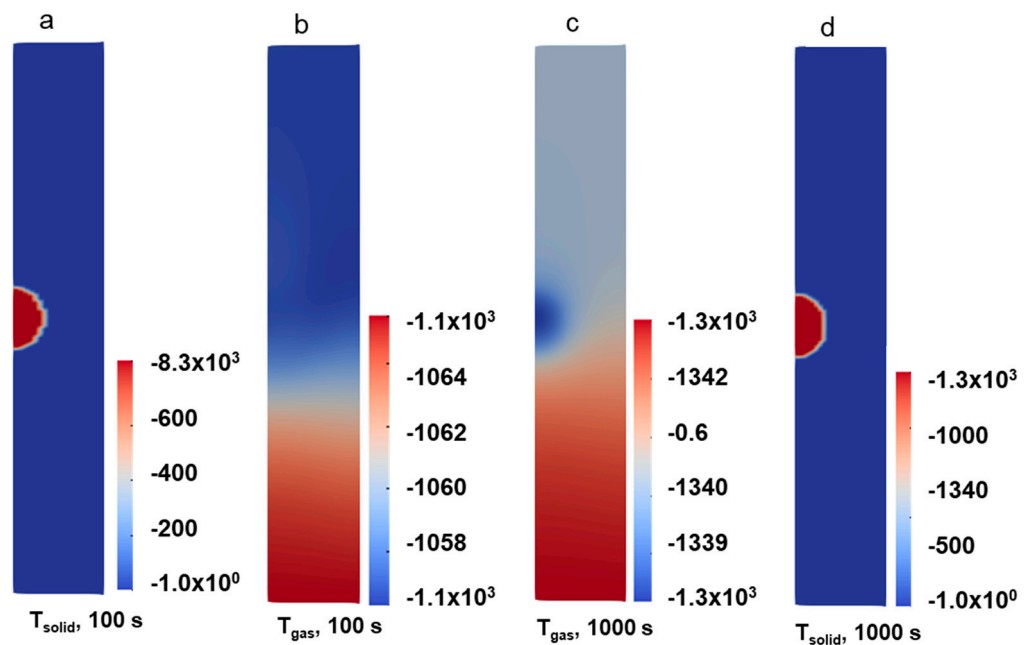

**Figure 11.** Gas phase (**b**,**d**) and Solid phase (**a**,**c**) temperatures at reduction times of 100 and 1000 s.

The Axial mass fraction profiles indicate that the gradients exist from the surface of the pellet to the center due to the reduction reactions. In addition, the axisymmetric effects can be observed from the mass fraction profiles of $H_2$ and $H_2O$ as shown in Figure 14. Due to the observed axisymmetric effects, the 2D axisymmetric geometry can be employed which results in significant reduction in the computational expense.

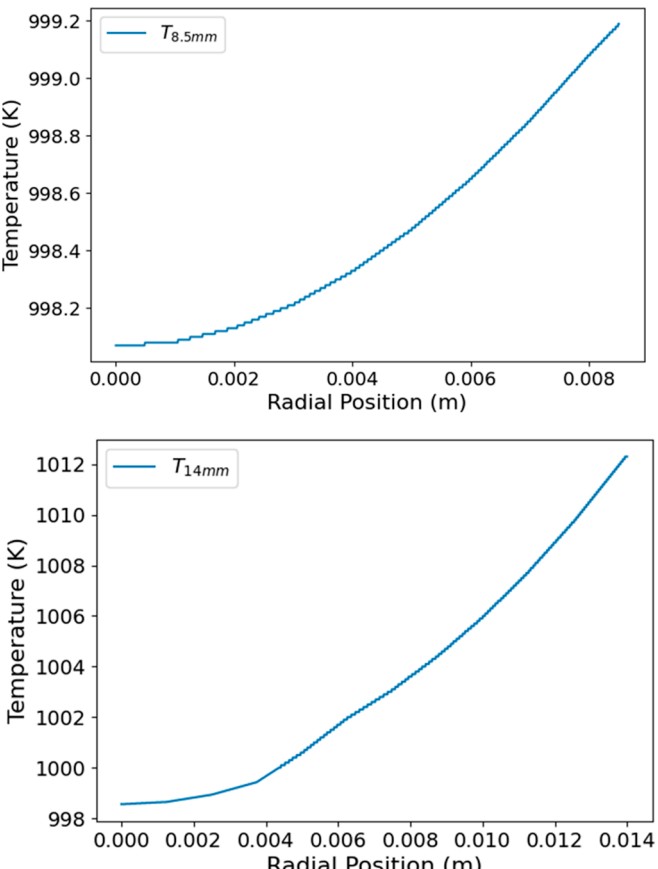

**Figure 12.** The radial temperature profiles in an 8.5 mm and 14 mm pellet.

### 5.2. Model Verification

The model results were compared with experimental data to assess its accuracy and generality. In the present study, the surface mass fractions were averaged over the arc length from each time step and compared with the experimental results. The surface mass fractions were chosen for model verification because diffusion controls the reaction rates, thus the surface and center mass fraction are different. The results in Figure 15 reveal that the model and experimental results are in close agreement. However, the average mass fraction across the entire pellet shows some deviation from the experimental results and this difference is attributed to slow diffusion of the reacting gas into the porous medium, thus resulting in slow reaction rates. The volume averaged mass fraction across the pellet was done as shown in Equation (19).

$$\overline{Y_{Fe}} = \frac{1}{A} \iiint Y_{Fe}(x, y, z) dx dy dz \tag{19}$$

### 5.3. The Effect of Pellet Size on the Degree of Metallization

The effect of pellet size on the degree of reduction was investigated. In order to isolate the influence of venturi effects on the reaction rates, the ratio of the diameter of the cylinder to the diameter of the pellet was kept constant as illustrated in Figure 16. The results in Figure 17 reveal that the reduction rate for a 4.5 mm, 5.5 mm and 7.5 mm pellet is the same, the difference is observed for a 14 mm pellet size. This can be attributed to the longer diffusion path length in the case of a 14 mm pellet.

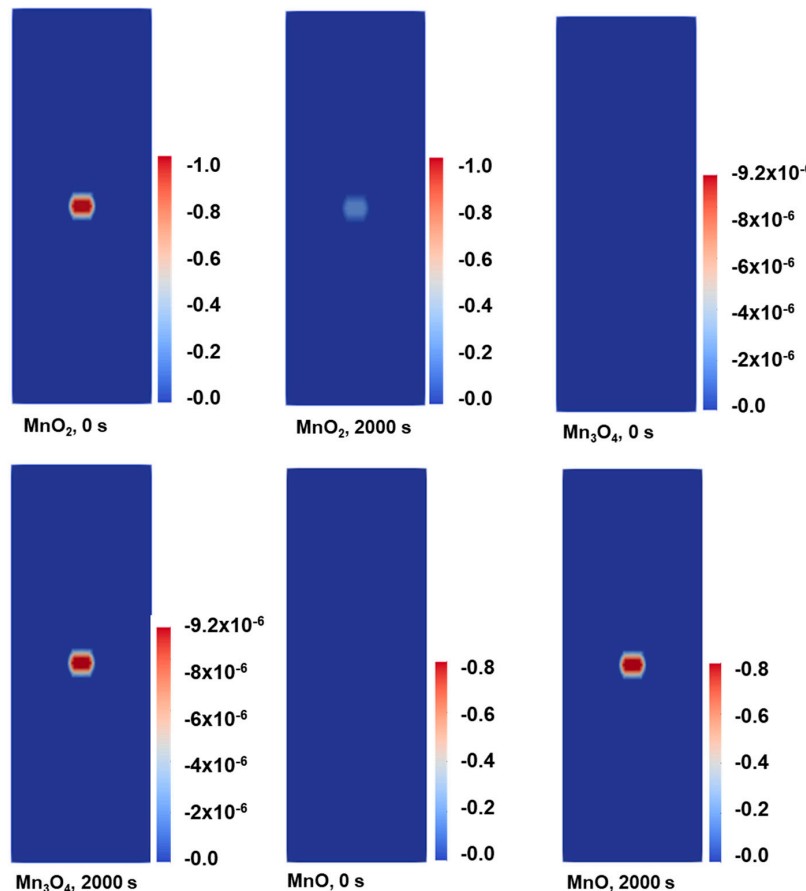

**Figure 13.** Mass fraction profiles of $MnO_2$, $Mn_3O_4$ and MnO at 0 and 2000 s reduction time.

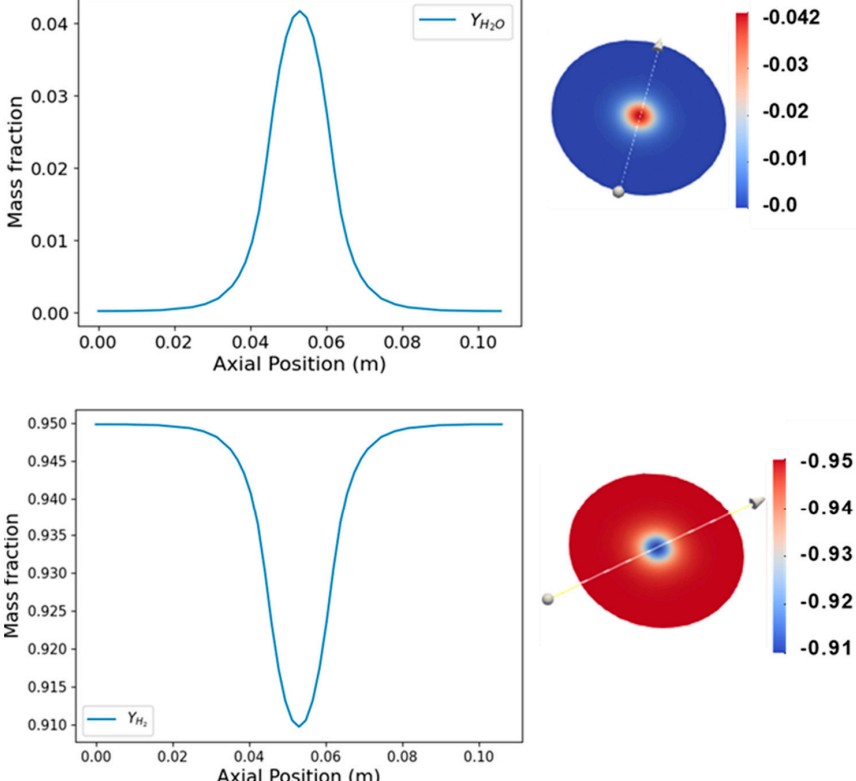

**Figure 14.** $H_2$ and $H_2O$ center-line mass fraction profiles.

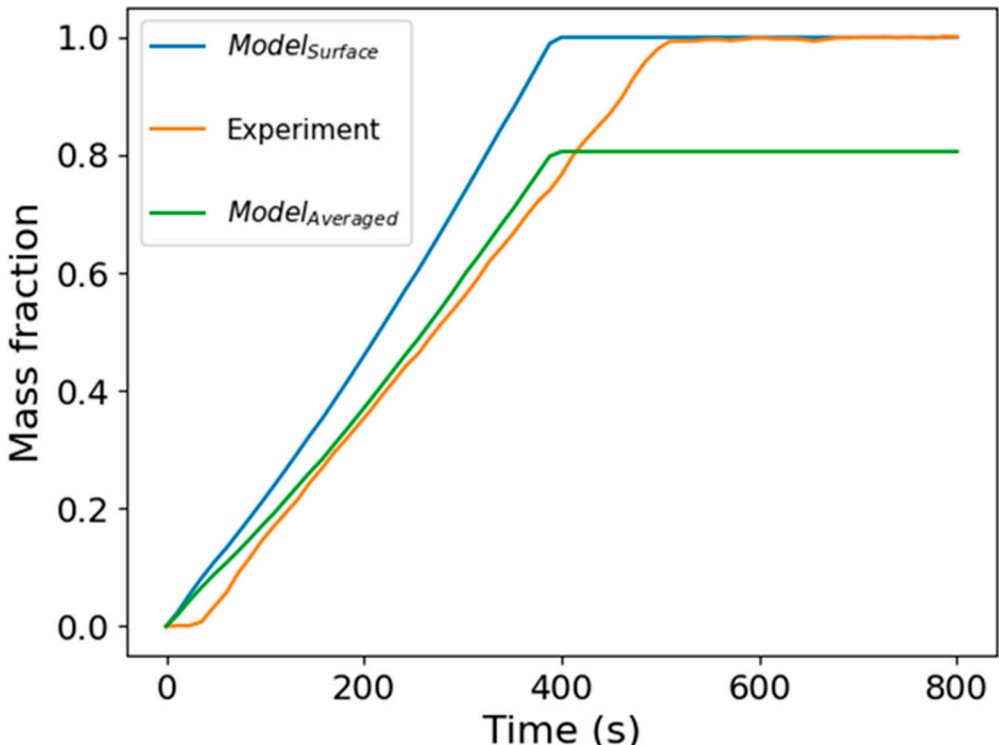

**Figure 15.** Model verification results for mass fraction of Fe in the reduction of $Fe_2O_3$ by $H_2$.

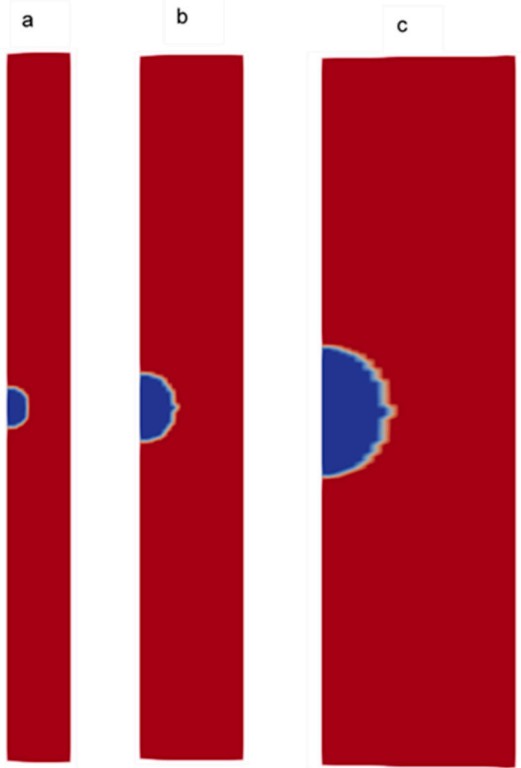

**Figure 16.** Pellets of various sizes (**a**) 5.5 mm pellet (**b**) 7.5 mm pellet and (**c**) 14 mm pellet.

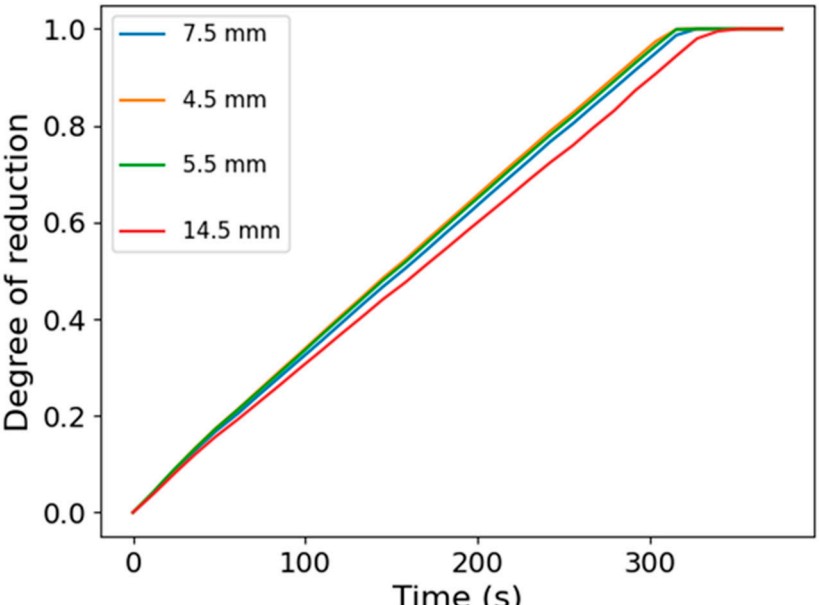

**Figure 17.** The degree of reduction at various pellet sizes.

## 6. Conclusions

The dynamic nature of iron and manganese oxides reduction with $H_2$ has been captured by CFD reacting flow solver. The formation of intermediate oxides and the final product as defined by the reaction mechanisms has been predicted with accuracy. The endothermic reactions in the porous medium render the assumption of local thermal equilibrium invalid as observed from the temperature profiles. There exist thermal and concentration gradients radially in spherical pellets and heat transfer between the gas and solid has to be taken into account. The high conductivity of iron ore pellets allows for an enhanced heating of the solid matrix to facilitate the reactions. Despite the enhanced heating to improve reaction kinetics, the diffusion limitations result in longer simulation times to achieve a complete metallization in a single pellet and for a macro scale approach, the computational expense will be higher. A semi-empirical model will be developed for a macro-scale model to allow for a faster evaluation of reaction rates and associated fields, thus aid the design and optimization of the pre-reduction systems.

**Author Contributions:** Conceptualization, M.K. and Q.R.; methodology, M.K.; software, M.K. and Q.R.; validation, M.K. and Q.R.; formal analysis, M.K.; investigation, M.K.; data curation, M.K.; writing—original draft preparation, M.K.; writing—review and editing, M.K.; visualization, M.K.; supervision, Q.R.; project administration, M.K. All authors have read and agreed to the published version of the manuscript.

**Funding:** This research received no external funding.

**Data Availability Statement:** The data that support the findings of the current study are available from the corresponding author upon request.

**Acknowledgments:** This paper is published by permission of Mintek. The authors acknowledge the Centre for High Performance Computing (CHPC), South Africa, for providing computational resources to this research project.

**Conflicts of Interest:** The authors declare no conflict of interest.

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
