# Peer review of "CFD Modelling of Gas-Solid Reactions: Analysis of Iron and Manganese Oxides Reduction with Hydrogen"

_mca, doi:10.3390/mca28020043_

Round 1
Reviewer 1 Report
The work performed CFD simulation of the very complex reduction process using H2. The SOA research limitation is very clear, the methodology is reasonable, and the results delivered are well discussed. Novel contributions are made towards accurate prediction of the process. Overall, The manuscript is very well written and can be published as it is.
Author Response
Thank you for the review.
Reviewer 2 Report
This paper presents an application of CFD modelling of gas-solid reactions. It is a topic of interest to the readers of JECE and the developed model along with some conclusions may be helpful for the researchers in the related areas. However, the paper needs very significant improvement before its acceptance for publication. My comments are as follows:
1、The figures in your paper are a bit blurry. Please consider replacing them with clearer ones.
2、There are some mistakes in text may be from conversion process of your manuscript. The authors should check.
3、The conclusion requires a more detailed discussion.
4、The equations need to be carefully examined.
Author Response
Reviewer’s comments: The figures in your paper are a bit blurry. Please consider replacing them with clearer one
Authors’ response: We agree that the pictures are blurry, and all the blurry pictures have been replaced with clearer pictures.
Reviewer's comments: There are some mistakes in text may be from conversion process of your manuscript. The authors should check.
Authors’ response: We agree with the reviewer that there are mistakes in text that need to be corrected. Below is a list of the corrections made
- On page 2 line 70, the grammatical mistake was corrected, “development in the use H2 as a reducing agent” was changed to “development of H2 as a reducing agent”.
- A correction in line 77 was made, “700oC” was corrected to “700 oC”. In line 79, “100%” was changed to “100 %”, the missing space was added.
- The extra spacing on line 82 and line 7 was removed. This was done for the rest of the document.
- A typographical error on line 86 was fixed “overned” was corrected to “governed.”
- On line 86, “taken account” was corrected to “taken into account.”
- In line 190, the correct symbol for density has now been used.
- In line 329, the error in punctuation has been corrected.
Reviewers’ comments: The conclusion requires a more detailed discussion.
Author’s response: We agree and have provided a more detailed discussion and we have revised the conclusions sections to add more details.
Reviewers’ comments: The equations need to be carefully examined.
Authors’ response: We agree that the equations needed to be examined and we have since made some corrections as stated below.
- The omission of Darcy’s resistance term in the momentum equation has been corrected in equation 7
- Equation 8 was also corrected, the summation to include all the gaseous species involved in the reaction has now been included.
- In Equation 9, ρs has be corrected to ρg since the equations described the gas phase mass balance.
- In Equation 10, C has been changed to Cp, there was an error of omission. On the same equation, the second term on the left-hand side of the equation has also been corrected. The third term on the left-hand side of equation 10 was also corrected.
- Equation 17 has also been corrected.
- Equations 1 to 6 were written with consistent spacing between variables.
